# Development of Bath Chemical Composition for Batch Hot-Dip Galvanizing—A Review

**DOI:** 10.3390/ma13184168

**Published:** 2020-09-19

**Authors:** Henryk Kania, Jacek Mendala, Jarosław Kozuba, Mariola Saternus

**Affiliations:** 1Department of Advanced Materials and Technology, Faculty of Engineering Materials, Silesian University of Technology, Krasińskiego 8, 40-019 Katowice, Poland; henryk.kania@polsl.pl; 2Department of Aviation Technologies, Faculty of Transport and Aviation Engineering, Silesian University of Technology, Krasińskiego 8, 40-019 Katowice, Poland; jacek.mendala@polsl.pl (J.M.); jaroslaw.kozuba@polsl.pl (J.K.); 3Department of Metallurgy and Recycling, Faculty of Engineering Materials, Silesian University of Technology, Krasińskiego 8, 40-019 Katowice, Poland

**Keywords:** hot-dip galvanizing, batch galvanizing, zinc coatings, galvanizing bath, corrosion resistance

## Abstract

Obtaining zinc coatings by the batch hot-dip galvanizing process currently represents one of the most effective and economical methods of protecting steel products and structures against corrosion. The batch hot-dip galvanizing process has been used for over 150 years, but for several decades, there has been a dynamic development of this technology, the purpose of which is to improve the efficiency of zinc use and reduce its consumption and improve the quality of the coating. The appropriate selection of the chemical composition of the galvanizing bath enables us to control the reactivity of steel, improve the drainage of liquid zinc from the product surface, and reduce the amount of waste, which directly affects the quality of the coating and the technology of the galvanizing process. For this purpose, the effect of many alloying additives to the zinc bath on the structure and thickness of the coating was tested. The article reviews the influence of various elements introduced into the bath individually and in different configurations, discusses the positive and negative effects of their influence on the galvanizing process. The current development in the field of the chemical composition of galvanizing baths is also presented and the best-used solutions for the selection and management of the chemical composition of the bath are indicated.

## 1. Introduction

Hot-dip galvanizing is one of the most effective methods of protecting steel against corrosion. It allows one to obtain good quality coatings, ensuring long-term protection with relatively low operating costs of the coatings. Despite the fact that the galvanizing process has been known for over 150 years [1], the continuous development of this technology is observed. The improvement of the process includes activities aimed at the rational use of zinc and energy in the process, the use of new technical solutions, but also the constantly expanding range of products that are protected with zinc coatings. The good properties of zinc coatings determine that nowadays not only small products and structures made of ordinary grade steel are galvanized, but also products made of high-strength steels after heat treatment [2] and wires intended for high-speed drawing [3]. This makes it necessary to conduct continuous research and solve problems with obtaining high-quality coatings. At the same time, rising zinc prices and decreasing zinc resources force the reduction in its consumption.

The cost of the coating depends on the amount of zinc used per unit of galvanized products. The zinc consumption depends not only on the thickness of the coating, but also on the losses resulting from the specificity of the technology, i.e., the formation of hard zinc and zinc ashes, oxidation of the bath surface, formation of solidified zinc droplets. Due to the rational conduct of the technological process, it is possible to reduce zinc consumption and limit its losses.

One of the most important factors determining zinc consumption is the chemical composition of the zinc bath [4,5,6,7,8,9]. The presence of alloying additives strongly influences the morphology [10,11], growth kinetics [12,13] and structure of the coating [14,15], as well as the properties of the bath itself [16]. Many years of research have allowed us to determine the impact of many elements introduced individually or in more complex configurations. Many of these solutions have been applied in industrial practice [6,8,9]. However, experience shows that it is currently impossible to indicate one best and universal chemical composition of the bath.

When selecting the chemical composition of the bath, one should be guided by the individual needs resulting from the specificity of the galvanized range of products. The main criteria for selecting bath additives are:
chemical composition of steel,generating zinc losses during the process,the shape, complexity and size of the product,intended use and application of the product,risk of product rupture in liquid zinc,coating resistance to corrosion.


The amount of zinc in the coating results from the reactivity of steel [17] in the liquid zinc and the ability of zinc to flow from the surface of the product when it is pulled out of the bath [18]. In many cases, especially when galvanizing steels with silicon content from the Sandelin series and high-silicon steels [19,20], the coatings are much thicker than allowed by the EN ISO 1461 standard [21]. The production in too thick coatings is economically unjustified, which leads to an increase in the consumption of zinc.

One of the wastes that causes zinc loss is galvanizing ash. These include oxides formed upon contact of liquid zinc with air [22]. Another problem is the removal of excessive amounts of zinc from the bath due to insufficient zinc discharge from the product surface. Many galvanized products have complex shapes and a high degree of surface development. Zinc accumulating in hard-to-reach places is pulled out with the product, irrationally increasing the consumption of zinc. In many cases, the zinc bath can affect the properties of the product itself, leading to its deformation and cracks [23]. The chemical composition of the bath may also affect the environment and human health [4,18] and determine the corrosion resistance of the coatings [24].

From a historical point of view, for many years of using the batch hot-dip galvanizing technology, the chemical composition of the bath was not selected or monitored in any particular way during the process. However, for nearly four decades, relatively intensive research [4,5,6,7,8,9] has been observed aimed at developing the best and universal chemical composition of the bath, which would allow the elimination or reduction in any unfavorable phenomena occurring during the hot-dip galvanizing process.

Therefore, the article presents an overview of possible solutions for the selection of the chemical composition of the galvanizing bath in relation to the changes that have occurred during several decades of intensive work on the development of the chemical composition of the batch hot-dip galvanizing bath. The influence of various alloy additions to the bath was analyzed depending on the way of their influence on the technological process and the quality of coatings. The analysis included contamination in the zinc galvanizing bath, additives reducing the amount of zinc ash, reducing the reactivity of Si-containing steels and additives improving the surface layer quality of coatings. The benefits and negative effects of their use were pointed out, and the best currently used solutions for the selection and management of the chemical composition of the bath were indicated.

## 2. Zinc Bath Impurities and Their Influence on the Galvanizing Process

Impurities in the zinc bath are those elements that are unintentionally introduced into the zinc bath as a result of the dissolving processes taking place or, together with the batch material, into the bath.

Good Ordinary Brand (GOB) zinc, commonly used for galvanizing in the 1980s, with lower purity 98.5 wt %, contained many impurities [25]. Their total content was up to 1.5 wt %, while the maximum Pb content was determined to be 1.4 wt %. The remaining impurities are Cd, Fe, Cu. This meant that the galvanizing bath always contained lead with content close to the solubility limit in liquid zinc of 1.2 wt % [26]. Furthermore, its excess, due to the much higher specific weight (11.3 g/m^3^) compared to zinc (7.13 g/cm^3^), formed a layer on the bottom of the bath. The presence of lead on the bottom of the bath has been found to be beneficial as it facilitates the removal of the hard zinc in the bath. In addition to Pb, the galvanizing bath contained Cd and Cu, which were never used as deliberately introduced alloying elements. Cd increases the fluidity of the zinc bath and improves the susceptibility of the coatings to plastic processing [27]. However, above 0.15% may increase the solubility of steel, which results in the formation of larger amounts of zinc in the bath [28]. Cadmium also has adverse effects on human health. On the other hand, copper adversely affects the adhesion of the coating to the substrate [29].

The production of 99.995% high-purity Special High Grade (SHG) zinc has increased significantly since the 1980s [25], where the level of all impurities does not exceed 0.005 wt % [30]. The consumption of lead-containing zinc alloys has decreased over the years and many galvanizing plants now use SHG zinc-based zinc baths. Of the impurities present in GOB zinc, only Pb became a deliberately added alloying additive to the bath. However, its presence in the bath is very controversial due to its negative impact on the environment and human health [4,18].

After eliminating the use of GOB zinc as a feedstock for the galvanizing bath, the only natural contamination in the bath is currently iron. Iron enters the zinc bath as a result of dissolving the substrate of galvanized products and dissolving the steel in the galvanizing bath, reaching the saturation state in liquid zinc. The limiting solubility of iron in zinc at a temperature of 450 °C is 0.03 wt %. On the other hand, in high-temperature galvanizing, it increases 10 times to 0.3 wt % at 550 °C [31]. Iron dissolved in the bath above the saturation level, depending on the temperature of the galvanizing process, releases the ζ phase in the form of particles in low-temperature galvanizing or δ_1_ phase particles in high-temperature galvanizing, creating the so-called hard zinc, which contaminates the galvanizing bath as waste.

The actual hot-dip galvanizing process is almost always carried out at the edge of the bath saturation with iron. The effect of the iron content in the bath on the formation of the coatings has not been thoroughly investigated. Harper and Browne [32] claim that under the conditions of high-temperature galvanization, the iron content in the bath increased to a saturation level of 0.3 wt % causes an almost two-fold increase in the thickness of the coating. Considering the industrial galvanizing process, in fact, there may be only slight fluctuations in the iron content below the saturation level as a result of the local lowering of the bath temperature and the supplementing the losses in the galvanizing bath by SHG zinc or its alloys. However, dissolving the iron in the bath near its saturation point causes the hard zinc to be released continuously. Its presence in the bath negatively affects the galvanizing process. The formation of hard zinc increases the unjustified consumption of zinc. The specific gravity of hard zinc is slightly higher than the specific gravity of liquid zinc. Therefore, hard zinc particles sink to the bottom of the bath, but its excessive amount in the bath hinders the galvanizing process. As a result of the extraction of hard zinc particles, coating defects in the form of thickenings and unevenness appear on the surface of the product. This requires the constant removal of the hard zinc from the bath. Therefore, while the iron is not a metal deliberately added to the bath, it should be considered a permanent component of the bath. This requires constant control of the sources of its penetration into the bath by selecting the optimal immersion time for galvanized products, maintaining an appropriate bath temperature and continuous control of flux purity.

## 3. Bath Additives for Limitation of Zinc Ash

Galvanizing ashes, apart from hard zinc, are the second main waste from the hot-dip galvanizing process, the formation of which is the source of an unjustified increase in zinc consumption.

The condition of the zinc bath surface affects the quality and aesthetics of the coating. The surface of liquid zinc oxidizes very quickly on contact with air. The resulting ZnO oxide allows oxygen to flow to the surface of the liquid metal, which means that the oxidation process is intense and continuous. The flux decomposition products also remain on the surface of the bath when the product is immersed. Both the ZnO formed and the flux decomposition products require continuous cleaning of the bath mirror. Contamination in the form of zinc ashes removed from the surface of the bath increases the consumption of zinc. A rationally carried out galvanizing process, therefore, requires limiting the amount of zinc ash.

In unit hot-dip galvanizing processes, aluminum additive has been used almost from the beginning of the industrial application of this technology. Al is mainly added to reduce oxidation of the bath surface. Due to the higher chemical affinity of aluminum for oxygen than for zinc, a continuous layer of Al_2_O_3_ is formed on the surface of the bath [22]. Due to its compact structure, this layer is a barrier to the inflow of oxygen to the surface of the liquid metal, thus protecting the bath surface against oxidation. Limiting oxidation on the surface of liquid zinc reduces the amount of zinc ash produced.

The effective reduction in the oxidation of the bath by Al is achieved at its concentration of 0.005 wt %. Al. However, the addition of Al in the bath reacts with the components of the flux to form the volatile AlCl_3_, a toxic white smoke. For this reason, the content of Al. in the bath should not exceed 0.01 wt % [33]. The moisture in the flux also enables the formation of Al_2_O_3_ on the steel surface. Under such conditions, coating discontinuities appear [34]. The appearance of the surface of the coating obtained in a bath containing an excessive amount of aluminum (0.25 wt % Al) on the chain, which was covered with a layer of flux before immersion in the bath, is shown in Figure 1. Very fine but numerous discontinuities can be observed in the coating. This is the typical surface appearance of a coating obtained in a bath containing an excessive amount of Al. It is common practice that the Al content in the galvanizing bath by the unit method does not exceed 0.01 wt %, in order not to adversely react Al with the flux.

Maintaining the appropriate concentration of Al in the zinc bath allows for a significant reduction in zinc losses and its consumption, while not damaging the coating.

## 4. Bath Additives for Control of Reactive Steel

### 4.1. The Reactivity of Steel in Liquid Zinc

The reactivity of steel in liquid zinc depends on its chemical composition. The most important alloying additive in steel, which intensely affects the galvanizing process, is silicon. Its presence in steels intended for galvanizing is mainly caused by the steel being killed with silicon (silicon-killed steel). Silicon-killed steels typically contain more than 0.03 wt % of Si. For this reason, efforts are being made to eliminate silicon from steel by steel killing with aluminum. Despite this, in industrial practice, steels with such an unfavorable Si content are very often encountered. More and more often, alloy steels are also being used for galvanizing, where the Si content determines the strength properties. In such steels, silicon removal is impossible due to the loss of the required properties of the steel.

Silicon in steel very intensively affects the thickness of the coating and its structure. In the case of galvanizing silicon-killed steels, the formation of the coating is unpredictable and the resulting coatings are very thick and show poor adhesion to the steel. This range of steel reactivity was investigated many years ago by Sandelin [19] and changes in coating properties due to Si interaction are called Sandelin’s effect.

The rate of reaction between iron and liquid zinc is very strongly dependent on the Si content in the steel. In terms of silicon content, steels are divided into low-silicon steels with a silicon content of less than 0.03%, steels from the Sandelin range containing 0.03–0.12% Si, steels from the Sebisty range containing 0.12–0.22% Si and high-silicon steels with a silicon content of more than 0.22%. The range of silicon concentration in steel determines the thickness of the coating and its structure, which determine not only the appearance and properties of the coating, but also the consumption of zinc.

In low-silicon steels, the coating thickness increases in accordance with the parabolic growth law [35]. The kinetics of coating growth on low-silicon steel shows that the extension of the immersion time does not significantly affect the thickness of the coating. Under such conditions, it is easy to control the thickness of the coating. This state changes quickly during galvanizing of Sandelin steel, where the coatings are several times thicker, and the thickness of the coating changes according to the linear growth curve [17]. For example, Sandelin steel containing 0.05 wt % Si, after immersion for 12 min, the thickness was 545 µm, while on low-silicon steel with the content of 0.02 wt % it was 88 µm (Figure 2a). Steels from the Sebisty range show lower reactivity in liquid zinc and a return to the parabolic law of increasing the thickness of the coating. In contrast, high-silicon steels again show a sharp increase in the thickness of the coating according to the linear law. For example, Sebisty steels with 0.18 wt % Si and high silicon steel with 0.32 wt % coating thicknesses of 221 µm and 488 µm, respectively, were obtained after an immersion time of 12 min (Figure 2a). The thicknesses of these coatings are smaller than that of the Sandelin range steels, but at the same time, they are much thicker than those of low-silicon steels. This means that after exceeding the content of 0.03 wt % Si in steel, excessively thick coatings are formed. Under industrial process conditions, the only real parameter that allows for coating growth control is the time of immersion in the zinc bath. As shown in Figure 2b, shortening the immersion time reduces the sensitivity of the reactive steels to the action of liquid zinc. However, with large dimensions of galvanized products, and the inability to control the chemical composition of steel on finished products, it is practically impossible to select an appropriate immersion time on reactive steels.

The change in the steel reactivity in liquid zinc changes not only the thickness of the coating but also its structure. The coating obtained on low-silicon steel (Figure 3a) shows a layered structure. The diffusion zone is formed by layers of the intermetallic phases of the Fe-Zn system: Γ, δ_1_, ζ. The phase characteristics of the Fe-Zn system are presented in Table 1. The Γ phase layer is difficult to observe because it has a small thickness, usually not exceeding 1 µm [36]. The δ_1_ phase layer is much thicker; its thickness is uniform and its structure is relatively compact. The next ζ phase layer shows two distinct zones. The inner zone with a clearly compact morphology turns into a heterogeneous structure in the outer zone. Such a two-zone structure of ζ phase results from the mechanism of its formation. The compact zone is formed by reactive diffusion between Fe and Zn, which leads to the growth of the layer. However, this layer is in direct contact with the liquid zinc which leads to its dissolution in liquid zinc. As a further consequence, in the zinc saturated with iron, the ζ phase is secondarily separated, creating a zone of loosely packed elongated crystals [20]. The diffusion layer of the coating formed in the zinc bath becomes covered when the product is taken out of the bath with the outer layer of the iron–zinc solution η. It is believed that the structure of coating shaped in this way is correct, making it possible to obtain the required and easily controlled thickness.

However, increasing the Si content in steel changes this structure. On steel from the Sandelin range (Figure 3b), the coating has a significantly developed diffusion layer. In the structure, a significant increase in the thickness of the ζ phase layer can be observed, which has a compact structure and a uniform thickness. At the same time, with increasing the thickness of the ζ layer, the thickness of the δ_1_ phase is significantly reduced. The diffusion layer of the coating extends almost to the surface of the coating, and the η outer layer is very thin or not present at all. The structure of the coating changes in the Sebisty range (Figure 3c). It is similar to the structure of the coating on low-silicon steel, although the morphology of the ζ phase is less compact. With higher silicon content in high-silicon steels (Figure 3d), the coating has a multiphase structure formed by Fe-Zn intermetallic phases, but no clear layered structure. The interaction of Si, which does not dissolve in the intermetallic phases of the Fe-Zn system, changes the morphology of the ζ phase, which forms loosely spaced crystals. This allows for the direct contact of the δ_1_ phase with the liquid zinc. As a result, phase δ_1_ begins to grow, creating a two-phase area δ_1_ + η, which causes an additional increase in the thickness of the coating.

In practice, all coatings obtained on steels containing more than 0.03 wt % Si have a much greater thickness than required. This contributes to an unjustified increase in zinc consumption. Moreover, the greater share in the structure of brittle intermetallic phases of the Fe-Zn system reduces the adhesion of the coatings to steel, and thus the resistance of the coatings to mechanical damage. Due to this, research has been carried out for many years to reduce the reactivity of steel in liquid zinc, and the mainstream research is focused on the study of the effect of various zinc bath additives on coating growth.

### 4.2. Effect of Nickel

The addition of nickel to the galvanizing bath is now widely accepted as a way to reduce the reactivity of Si-containing steels in liquid zinc. The first commercial applications of nickel additive began in 1982-then 0.13 wt % Ni was added to the bath. During the first 20 years of industrial use of the Ni additive in the bath, a number of changes were introduced and many studies on this subject have been published. Nickel as a pure metal dissolves relatively slowly in liquid zinc. Hence, it was initially added to the bath in the form of Zn-2%Ni mortar. In order to increase the efficiency of equalizing the Ni content in the bath volume, it was also introduced into the bath in the form of a powder with intensive stirring. Currently, Ni is most often added as the Zn-0.5% Ni alloy. At that time, its content in the bath also changed. Even in the 1990s of the last century, its content in the bath decreased to 0.9 wt %. Finally, the optimal range of Ni concentration in the zinc bath used so far was determined as 0.04 wt % to 0.06 wt % [5,6]. Nickel in this content range enables the suppression of Sandelin peak, and the coating on Sandelin steel obtained in such a bath has a structure similar to that of the coating obtained on low-silicon steel (Figure 4a). The presence of nickel in the bath does not change the structure of Sebisty steel (Figure 4b) and high-silicon steel (Figure 4c).

There is also a clear reduction in the thickness of the coating on Sandelin steel (0.05 wt % Si), while on Sebisty steel (0.18 wt % Si) and high-silicon steel (0.27 and 0.32 wt % Si), the reduction in the thickness of the coating is insignificant (Figure 5a). The higher Ni contents in the bath, in addition to higher costs, may adversely affect the galvanizing process and the quality of the coating. Excessive Ni content in the bath causes floating dross, which contaminates the bath [41]. Floating dross is formed by fine particles of the Γ_2_ phase described by the Fe_6_Ni_5_Zn_89_ formula, which is isostructural with the Γ phase of the Fe-Zn binary system [42]. In the Fe-Ni-Zn triple equilibrium system [43], the Γ_2_ phase is stable at Ni content over 0.06 wt %. The Γ_2_ phase particles have a regular shape and separate from the Zn + Ni bath supersaturated with iron at the border with the ζ phase layer. In such conditions, the floating dross is pulled out with the product from the bath, creating defects on the surface of the coating and increasing the amount of zinc withdrawn. The formation of the Γ_2_ phase also entails a faster depletion of Ni from the bath. In such a situation, increasing the Ni content in the bath above 0.06 wt % becomes unreasonable.

Higher Ni contents cause a significant increase in the thickness of the coating. Simultaneously with the increase in the Si content in the steel, the thickness of the coating slightly decreases (Figure 5a). The coating structure also changes. Higher Ni content changes the equilibrium in the Fe-Ni-Zn system. The δ_1_ phase is in equilibrium with liquid zinc [44]. Direct contact of the δ_1_ phase with liquid zinc causes its growth. In the structure of the coating, the occurrence of the two-phase zone δ_1_ + η can be observed (Figure 5b). The ζ phase does not form a continuous layer, but single, loose crystals at the boundary with the δ1 phase [45].

Nickel, along with aluminum, is now the standard alloying additive introduced into the zinc bath to reduce the reactivity of the steel. The vast majority of galvanizing plants use a Ni additive in combination with other alloying additives improving the efficiency of the galvanizing process.

### 4.3. Effect of Aluminium

Aluminum in amounts added to traditional zinc baths (up to 0.01 wt %) does not affect the structure of the coating. Higher Al contents in liquid zinc lead to a change in the equilibrium conditions in the Fe-Zn-Al system. In such a case, the Fe-Al phases are in equilibrium with liquid zinc instead of Fe-Zn phases [46]. The ζ phase is in equilibrium with the liquid zinc if the Al content in the liquid zinc does not exceed 0.1 wt %. Above 0.1 wt % there is δ_1_ phase in equilibrium with liquid zinc. Above 0.11 wt % Al in equilibrium with liquid zinc is the Γ phase. However, exceeding the content of 0.133 wt % Al, completely new conditions arise for the equilibrium of liquid zinc with the Fe_2_Al_5_(Zn) phase.

Under the conditions of the galvanizing process with the Al content in the bath above 0.133 wt %, the Fe-Al system phases are first formed. Studies have shown that a thin layer of Fe_2_Al_5_ and/or FeAl_3_ phases may develop on the steel surface [47]. However, this condition is unstable. The layer of the Fe-Al system phase disintegrates, which enables further growth of the Fe-Zn system phases. However, the initial formation of the Fe-Al system phases delays the growth of the Fe-Zn intermetallic phases. This interaction is commonly used in the Sendzimir method of continuous sheet galvanizing. Increased Al content allows us to reduce the thickness of the brittle intermetallic phases of the Fe-Zn system, thanks to which the galvanized sheet shows better susceptibility to plastic processing. This method does not use surface fluxing of the products prior to immersion in liquid zinc. There is therefore no risk of aluminum interacting with flux components.

In the batch hot-dip method, attempts were made to control the reactivity of the steel by increasing the Al content in the bath. It was argued that the reduction in steel reactivity could be achieved with the content of 0.1–1 wt % Al in a bath [48,49]. Gutman [48] found that silicon modifies the delay layer from the double phase of the Fe-Al system to the triple-phase of the Fe-Al-Si system, which enhances the delay effect. He also found that 0.05 wt % Al can delay the growth of the Fe-Zn system phases on high-silicon steels (~0.4% Si).

The production used a bath called Polygalva containing an addition of 0.035–0.045 wt % Al and 0.003–0.005 wt % Mg [50]. However, it is likely that the difficulties in carrying out the process with the unit method with a higher Al content caused the withdrawal of this bath from industrial production. Recently, the use of Al additive has once again been proposed to control the reactivity of steel. The increased Al content in the unit hot-dip galvanizing bath is to limit the coating thickness in a wide range of silicon concentration also on high-silicon steels. The authors of such solutions [7,51] suggest increasing the Al content to 0.05 wt %, while recommending the use of a modified flux. Although such a solution is used in industrial practice, the literature lacks detailed data on the effects obtained.

Research on the effect of bath additives [52] shows that the Al content may limit the effect of silicon on the thickness of the coating. Al addition at the level of 0.022 wt % for the bath containing nickel at a concentration of 0.04–0.06 wt % reduces the thickness of the coating, regardless of the Si content in the steel (Figure 6). However, in the case of steel from the Sebisty range and high-silicon steel, this does not change the character of the curve of growth, and the obtained coatings still have excessive thickness [52]. In the Sandelin steel coatings, this effect of nickel causes the thickness of the coatings to be comparable to the thickness of the coating obtained on low-silicon steel. However, the increased content of Al does not change the structure of the coating. The coatings obtained on the Sandelin steel in the Zn + Ni and Zn + AlNi baths (Figure 7) showed a similar layer structure: Γ, δ_1_ and ζ. However, a reduction in the thickness of the layer ζ of the coating obtained in the bath containing Al can be observed.

However, it should be emphasized that when using the unit method, increasing the Al content in the bath above 0.01 wt % is risky as it can react with the flux, leading to coating discontinuities.

### 4.4. Effect of Titanium and Vanadium

Titanium and vanadium are alloying additives that, when added to the zinc bath, can reduce the reactivity of the steel in a manner similar to that of Ni. Sebisty and Palmer [54] suggest that the addition of 0.01 wt % Ti to the bath slightly reduces the growth of the ζ phase on low-silicon steels. Reumont et al. [55] claim, however, that the interaction of Ti is similar to that of Ni due to the high similarity of the three-component equilibrium systems Fe-Zn-Ti and Fe-Zn-Ni. Research conducted in this area [55] confirmed the reduction in steel reactivity in baths containing 0.33–0.37 wt % Ti. The thickness and morphology of the coatings obtained on steels with a Si content of up to 0.167 wt % were similar to the thickness and morphology of the coatings on low-silicon steel. On the other hand, the reduction in the thickness of the coating on high-silicon steel (0.367 wt % Si) did not exceed 15%. With a lower content of 0.034 wt % Ti authors [55] found no decrease in the reactivity of steel in liquid zinc. It can therefore be argued that the addition of Ti to the bath compared to the addition of Ni also allows the reactivity of Sebisty’s steel to be reduced, but a much higher concentration in the bath is required. Reumont et al. [55] also observed the presence of large precipitates of the Γ_2_-Fe_2_TiZn_22_ phase in the structure of the coatings obtained in the baths with the addition of Ti, and also the precipitations of the TiZn_15_ phase with higher Ti contents. The analysis of the Fe-Zn-Ti equilibrium system at the temperature of 450 °C shows that in the bath supersaturated with Fe (0.035 wt %) with Ti content above 0.034 wt % in equilibrium with liquid zinc there is the Γ_2_ phase, while at the content above 0.34 wt % by the TiZn_15_ phase is also in equilibrium with the liquid zinc [56]. The addition of Ti to the bath reduces the reactivity of the steel, but the required content in the bath causes the formation of Γ_2_ and TiZn_15_ phase precipitates, which negatively affect the appearance of the coating, and at the same time intensely deplete the titanium in the bath.

The negative effects of Ti are eliminated by the addition of vanadium. Adams et al. [57] indicated that the addition of 0.04 wt % V plus 0.05 wt % Ti is effective in reducing coating growth on reactive steels to an extent similar to the addition of Ni to the bath. However, vanadium salts are very toxic, which limits their use [58].

### 4.5. Effect of Lead, Bismuth and Tin

The main reason for adding lead, bismuth and tin to the zinc bath is their influence on the shaping of the outer layer and the smoothness of the coating. To some extent, these metals can also contribute to the growth of the diffusion layer of the coating.

Lead has no major influence on the formation of the diffusion layer of the coating. Although Krepski [16] reports that lead reduces the diffusion layer of the coating, the studies by Sebisty and Edwards [59] show that low-carbon steel Pb has no effect on the growth of the Fe-Zn intermetallic phase layer. Additionally, Reumont and Perrot [18] report that the addition of lead to the bath does not change the morphology, phase composition of the coating and the growth kinetics of the intermetallic phase layers of the Fe-Zn system.

The addition of Bi and Sn also did not affect the structure of the diffusion layer of coatings obtained on low-silicon steels. Pistofidis et al. showed that the coating obtained in the bath containing up to 2% Bi has a typical layered structure of phases Γ, δ_1_ and ζ [14]. On the other hand, Katiforis et al. [29] claim that the addition of tin up to 3 wt %. does not affect the morphology of the coatings. Gilles et al. [60] showed, however, that with a content of at least 2.5 wt % in the bath, tin reduces the effect of Si in the steel. Additionally, Avettand-Fènoël et al. [15] found a reduction in coating thickness on Sandelin’s and hyper-Sandelin’s steel in baths containing 1 and 5 wt % Sn. The proposed mechanism of limiting the reactivity of steel assumes that due to the lack of solubility of Sn in the intermetallic phases of the Fe-Zn system, tin is released during the diffusion layer growth and due to the low surface tension it forms a thin barrier layer at the border of the diffusion layer with the bath, which probably hinders iron and zinc diffusion [60]. A similar mechanism of interaction of the barrier layer containing Sn and Bi was presented by Beguin et al. [4] in a Galveco bath containing both Sn and Bi.

Tin addition to a zinc bath in the range up to 2 wt % does not affect the coating thickness on Armco iron and low-silicon steels, but it reduces the coating thickness on reactive steels (Figure 8). On Sandelin range steel, a clear reduction in the thickness of the coating is visible at a content of 1.5 wt %; while on Sebisty’s steel and high-silicon steel, the reduction in the coating thickness increases in proportion to the increase in the tin content in the bath [61]. The addition of tin to the bath does not eliminate the Sandelin effect, but it significantly reduces it. Comparing the structure of the coating obtained on steel from the Sandelin range in a bath with the addition of 2 wt % of tin (Figure 9a) with the coating obtained in the “pure” zinc bath (Figure 9b), it is possible to observe a decrease in the thickness of the ζ layer with a simultaneous increase in the thickness of the δ_1_ phase. This is a typical phenomenon that indicates the inhibition of the coating growth process. As a result of the acceleration of diffusion processes on steels from the Sandelin range, in the extreme case, the coating during slow cooling after taking it out of the bath may crack and delaminate from the substrate (Figure 9c). In this case, the coating loses its protective properties. The addition of tin, although it does not completely eliminate the Sandelin effect, allows for protection against this extreme case [62].

The dissolution of the ζ phase in direct contact with liquid zinc is a diffusion process. Diffusion dissolution phenomena intensify in the case of increased reactivity in the Fe-Zn system caused by the presence of silicon. The diffusion coefficient of iron in a zinc bath depends on the concentration of iron in individual intermetallic phases of the Fe-Zn system and has the highest value for the dissolution process of Sandelin’s steel [64]. Diffusion dissolution tests of steel with different silicon content in a bath containing 2 wt % Sn [65] and 0.5 wt %. Bi [66] showed a reduction in the thickness of the layer supersaturated with dissolved iron on the reactive steels compared to the pure zinc bath. The greatest decrease in the thickness of the layer is observed on Sandelin’s steel, but Sn and Bi also reduce the thickness of the layer in high-silicon steels. In the baths containing Sn and Bi, the value of the diffusion coefficient D = f(c_Fe_) in the ranges of the intermetallic phases of the Fe-Zn system also decreases. The value of the diffusion coefficient decreases the most in the occurrence of the ζ phase on Sandelin’s steel. With the content of 6.5 wt % Fe in phase ζ the diffusion coefficient decreases from 220.31 × 10^−8^ cm^2^·s^−1^ in a pure zinc bath to 119.89 × 10^−8^ cm^2^·s^−1^ [66] in a bath containing 0.5 wt % Bi and 44.56.31 × 10^−8^ cm^2^·s^−1^ [65].

The content of Bi and Sn in the zinc bath also reduces the value of the δ_1_ phase diffusion coefficient. The dependence of the diffusion coefficient on the concentration of iron in the range of its concentration in the δ_1_ phase during the dissolution of Sandelin’s steel is shown in Figure 10 [53]. Thus, it can be argued that the presence of Sn and Bi additives in the bath has a positive effect on reducing the reactivity of steel in liquid zinc.

## 5. Bath Additives for Surface Layer Quality

The outer layer of the coating plays a protective role in the initial stage of corrosion. The thickness of the outer layer, its homogeneity and surface appearance determine not only the quality but also the aesthetics of the coating. These properties of the coating depend primarily on the fluidity of the zinc bath, surface tension and zinc drainage from the surface of the product, surface oxidation and the method of crystallization of the outer layer of the coating. These properties are shaped by the appropriate selection of alloying additives for the bath, which at the same time improve the to improve the manufacturability of the process and reduce the consumption and losses of zinc. The most important alloying elements used for this purpose are aluminum, lead, bismuth and tin.

Aluminum is not an alloying additive whose primary purpose is to influence the surface quality of the coating. Taking into account the previously described effect of this alloying additive, it should be noted, however, that the presence of Al in the bath also brightens the coating surface and improves its gloss immediately after its production. Due to the greater affinity of Al for oxygen, a thin layer of Al_2_O_3_ may also be formed on the surface of the coating, which protects the surface of the coating against corrosion immediately after its creation. The aluminum in the bath has little effect on the bath fluidity and surface tension, but it is believed that it may have an effect on the size of the spangle if metals such as lead, bismuth and tin are present in the bath. However, the test results do not support this property.

Lead lowers the surface tension of liquid zinc and improves the fluidity of the bath. This results in better zinc drainage from the surface of the product. Increasing the Pb content from 0.03 wt % up to 1.2 wt % causes a reduction of up to 60% of the amount of zinc drawn from the bath with the product [16]. Krępski [16] also showed that the zinc bath achieves the best fluidity at the content of 0.4–0.5 wt % Pb. However, a further increase in the Pb content in the bath reduces the fluidity of the zinc bath. The increase in the lead content in the zinc bath reduces the surface tension from greater than 760 dynes/cm for a pure zinc bath to about 530 dynes/cm in a bath containing 0.9 wt % Pb. According to Fasoyino and Weinberg [10] and Strutzenberge and Faderl [11], the reduction in surface tension is caused by lead segregation at the front of the crystallization of zinc dendrites. This also influences the formation of a spangle on the surface of the coating.

However, lead is toxic and harmful to both human health and the environment. In the EU and the USA, its use as an alloying additive in a zinc bath is limited in some cases [4,54]. The addition of Bi and Sn has become an alternative to the addition of lead. However, the favorable influence of lead on the galvanizing process and the quality of the coating mean that this metal is still relatively often added to the zinc bath.

Bismuth has several beneficial properties. It increases the castability of zinc, reduces the surface tension and is not harmful to the environment. Gagne [67] showed that the addition of 0.1 wt % Bi gives a similar intensity of liquid zinc flowing from the product surface as approx. 1 wt % Pb [67]. Additionally, the addition of 0.05–0.1 wt % Bi in the bath reduces the surface tension of the liquid zinc alloy from about 750 mJ/m^2^ to 620–650 mJ/m^2^ (dyn/m^2^). Such an effect of lowering the surface tension can be obtained with the concentration of lead in the bath at the level of 0.4–0.5 wt % [58]. Therefore, it is assumed that to achieve similar bath properties caused by the addition of Bi, its content is almost 10 times lower than that of the Pb addition. Bismuth also promotes the formation of a spangle on the surface of the coating [22,67].

Tin, although it is able to control the coating growth on reactive steels, is added to the zinc bath mainly to improve the quality and appearance of the coating. Tin as an independent alloying additive does not have a significant effect on the intensity of zinc drainage from the product surface, as well as the smoothness of the coating. Contrary to lead and bismuth, tin forms a spangle with a much higher content in the bath above 1 wt % (Figure 11); however, spangle caused by the presence of tin in the bath is much finer [36]. In combination with spangle-producing additives, the presence of tin in the outer layer of the coating causes changes in its solidification. This leads to a larger spangle with a distinct dendritic morphology—such appearance of the coating is more attractive and aesthetic for customers of galvanized products.

Lead, bismuth and tin, in addition to improving the surface quality of the zinc coating, also improve the productivity of the galvanizing process and reduce zinc losses. As a result of better zinc removal, fewer stains and icicles are formed on the surface of the product. This way, less zinc is lost when it is removed, and the cleaning time for the product itself is also reduced. Additives with a low melting point, improving zinc drainage, also allow the bath temperature to be slightly lowered. This has the effect of extending the service life of the galvanizing kettle and reducing energy costs.

The alloying elements Pb, Bi and Sn in addition to Al and Ni are therefore nowadays a constant component of the bath and are used interchangeably. The use of multi-component baths containing Bi and Sn, however, has encountered new problems related to the increasingly common cracking of steel structures during galvanizing. The research conducted in this area allowed the identification of the phenomenon of Liquid Metal Embrittlement (LME) [68], and the content of Bi [69] and Sn [70] in the bath was indicated as one of the main reasons for the destructive impact of the galvanizing process on steel structures. Hence, it is recommended to limit the Bi and Sn content to 0.1 wt % Sn and total Pb + 10Bi addition of less than 1.5 wt % [71]. Due to the much less stringent limitations of Pb, it has become a relatively popular alloying additive despite its harmful impact on the environment and health.

## 6. Alloying Zinc Bath for Batch Hot-Dip Galvanizing

The traditional galvanizing bath, which was made on the basis of GOB zinc, contained lead at the saturation level and slight contamination of other metals. In the initial stage of use, the bath was also saturated with iron. The first alloying elements introduced into the bath were small amounts of aluminum and tin. Their main task was to lighten the surface of the coating. This alloy was widely used for galvanizing until the 1970s of the last century, and the chemical composition and the content of alloying additives in the bath were practically not subject to any control. The needs of the last few decades, including the need to reduce the consumption of zinc, have resulted in the creation of many ready-made alloys for hot-dip galvanizing. The purpose of their use was to become independent of the chemical composition of steel, to improve the manufacturability of the galvanizing process and to increase the aesthetics of the zinc coating. Ultimately, however, the zinc alloys were to ensure a reduction in the amount of zinc loss during the process and thus in its consumption.

### 6.1. Commercial Zinc Alloys for Hot-Dip Galvanizing

Already in the 1970s, the first alloy bath, Polygalva, was developed. The initial alloy contained additives of 0.035–0.045 wt % Al, approx. 0.15 wt % Sn and 0.003–0.005 wt % Mg [72]. However, probably difficulties in carrying out the process with the batch hot-dip method with a higher Al content caused the withdrawal of this bath from industrial production. The development of Technigalva alloy initiated a new direction in the development of hot-dip galvanizing technology. The Technigalva alloy initially contained 0.13 wt % Ni [6], and its effectiveness in reducing the Sandelin effect resulted in the relatively widespread use of this alloy in the 1990s. At that time, there was a dynamic development of research into the development of new, improved alloys for galvanizing, which were to improve the manufacturability of the process and at the same time eliminate harmful lead. The market has launched such alloys as the Canada-developed BritePlus (Zn-Bi-Sn-Al) [60] and Galveco (Zn-Bi-Sn-Ni) [4] in Europe.

The BritePlus alloy contained up to 0.4 wt % Sn and up to 0.1 wt % Bi. The additives were to brighten and improve the gloss of the coatings and supplemented the lead content in the bath not exceeding 0.6 wt %. Additionally, nickel from 0.4 to 0.6 wt % and aluminum, the content of which did not exceed 0.005 wt % [8] were introduced into such a bath.

In 2000, the Galveco alloy was introduced to the market, containing significant amounts of tin from 0.9 to 1.2 wt %, bismuth from 0.08 to 0.1 wt % and nickel from 0.04 to 0.05 wt %. The bath based on this alloy was enriched with the addition of Al with contents of 0.005 to 0.01 wt %. The high tin content in the bath allowed to limit the reactivity of steel in the whole range of silicon concentration, which was additionally supported by the synergistic interaction with Ni [4]. However, the Galveco alloy [9], dynamically introduced into industrial practice, was finally withdrawn from use in 2007. At that time, the use of BritePlus [73] was also discontinued. The use of baths containing Bi and Sn made the cases of cracking of steel structures due to the LME phenomenon more common. This was the main reason why these alloys were withdrawn from production.

The galvanizing alloys showed a number of advantages, but did not guarantee the stability of the chemical composition of the bath. Pb, Bi and Sn are very stable bath alloying additives and their consumption is proportional to the amount of zinc removed. However, Al and Ni depleted from the bath in a manner disproportionate to the amount of galvanized products. Thus, the use of a finished alloy did not guarantee the optimal content of alloying additives in the bath during its operation.

### 6.2. Optimizing Zinc Bath Chemistry

Currently, the most rational method of managing the chemical composition of the zinc bath is the appropriate selection of the configuration of the bath additives and constant maintenance of their content in the bath within a narrow range of concentration changes that allow for their most effective interaction. The optimal chemical composition of the bath should contain alloying additives that will ensure the reduction in the amount of zinc ash, limiting the reactivity of the steel in the bath and improving the drainage of liquid zinc from the product surface. All these requirements are met by the group of five metals: Al, Ni, Pb, Bi and Sn, which are currently used as alloying additives to the zinc bath. The qualitative composition of the bath is selected individually and depends mainly on the range of products galvanized in a given galvanizing plant. Al and Ni are now permanent alloying additives to the zinc bath. On the other hand, Pb, Bi and Sn are used interchangeably, and their presence in the bath is determined by such criteria as the impact on the environment (Pb) and the risk of LME (Bi, Sn). Therefore, the optimal chemical composition of the bath in different galvanizing plants may be different. However, to ensure high coating quality and process efficiency, it is important to follow good bath management practices and to constantly control the content of the alloying additives in the bath. The optimal content of alloying additives in the bath should be:
for Al 0.005–0.01 wt %,for Ni 0.04–0.06 wt %,for Pb 0.4–0.5 wt %,for Bi 0.05–0.1 wt %,for Sn 0.1–0.3 wt %.


Analyzing the current knowledge on the influence of particular elements on the formation of zinc coatings, as well as on the technological properties of the bath, three alternative optimal chemical compositions of the bath based on the alloys: Zn-AlNiPb, Zn-AlNiBi and Zn-AlNiBiSn can be distinguished.

Figure 12 shows the structure of the coating obtained in the bath containing the optimal configurations and the concentration of Al, Ni Pb, Bi and Sn additives. The coating on low-silicon steel (Figure 12a,b) shows a typical layered structure of phases Γ, δ_1_, ζ and η. The coating obtained on Sandelin’s steel (Figure 12c,d) shows a similar phase morphology, which is due to the maintenance of an appropriate Ni content in the bath [12,13,24]. In the coating obtained in the lead-containing bath, the occurrence of a significant amount of Pb precipitates can be observed. Lead is released both on the coating surface (Figure 13a) and on the cross-section of the outer layer (Figure 13b). A privileged place for the release of lead is the boundaries of the zinc grains, but fine dispersion lead may be released at the grain cross-section. Despite the fact that the chemical composition of the bath ensures the correct structure and optimal conditions for the galvanizing process, the presence of lead precipitates may reduce the corrosion resistance of the coatings. The conducted tests of corrosion resistance of coatings obtained in the Zn-AlNiPb bath confirmed the reduction in the corrosion resistance of coatings in the environment of neutral salt spray and in a humid atmosphere containing SO_2_, compared to coatings obtained in the bath without alloying additives [24]. The reduction in corrosion resistance can be explained by the formation of local corrosion cells between the lead precipitates and the zinc matrix. Lead having a higher standard potential is a cathode in relation to Zn (*E°* (Pb^2+^ /Pb) = −0.1262 V, *E°* (Zn^2+^/Zn) = −0.7618 V; vs. SHE—Standard Hydrogen Electrode [74]). Similarly, the precipitation of Bi and Sn occurs in the outer layer (Figure 12b) and in the diffusion layer (Figure 13c) of the coating obtained in a bath containing these metals. Bi as well as Sn are more cathodic compared to Zn (*E°* (Bi^3+^/Bi) = 0,308 V, *E°* (Bi^+^/Bi) = 0,5 V, *E°* (Sn^+2^/Sn) = −0.1375 V vs. SHE [74]). Currently, there is no unambiguous information on the influence of Bi and Sn on the corrosion resistance of coatings, but when comparing the standard potentials, it can be concluded that the potential difference between the precipitates of Bi with a positive standard potential and Zn is greater than the potential difference between the precipitates of Pb and Zn. As a result, the corrosion action of the cell in which the cathode is Bi may be more effective compared to the corrosion cell in which the cathode is Pb.

The optimization of the chemical composition of the bath is currently focused on reducing zinc losses and reducing its unjustified consumption. Unfortunately, the influence of alloying additives on the corrosion resistance of coatings is almost completely ignored in the research.

### 6.3. Zinc Bath According DASt-Richtlinie 022

Since 2000, when the Galveco alloy was presented at the Intergalva conference in Berlin [4], baths containing Bi and Sn have been increasingly used in industrial practice. This allowed for the removal of harmful lead from the bath, while maintaining the beneficial technological properties of the bath, as well as controlling the reactivity of hyper-Sandelin’s steel. At that time, a significant increase in the cases of cracking of steel structures during galvanizing was also observed. The problem concerned mainly large welded load-bearing structures and steel products in places of strong plastic deformation. Figure 14a shows an example of a 6 m long welded support of a load-bearing structure, on which a 110 mm long crack was observed. The crack arose near the welded joint and its gap is not filled with zinc. In this case, it is possible to identify the crack before assembly. Much more dangerous are cases when a crack is masked by liquid zinc filling it. Figure 14b shows the clamp for the holder of a vortex pole with a diameter of 200–240 mm used in the power industry, where the crack was revealed only during the product’s operation. The view of the fracture indicates that it is covered with zinc, which indicates the crack initiation already during the galvanizing process. In this case, the characteristic location of the crack is the zone of intense plastic deformation. In most cases, the identification of cracks is very difficult and becomes apparent during the operation of the products, which is a direct threat to life.

Intensive research conducted in this area allowed researchers to indicate the LME phenomenon [23,68] as the cause of cracking. The occurrence of the LME phenomenon during galvanizing was known before, but it mainly concerned cracking in the galvanizing baths. Rädeker [75,76] claims that lead, regardless of the temperature, and bismuth up to the temperature of 300 °C do not cause cracking of the steel. However, the zinc-lead alloy at 450 °C favors the LME phenomenon. According to Tasak [77], in order to avoid the LME phenomenon, the start-up of a new bathtub should be carried out in pure zinc, and lead should be added after the zinc has melted and the temperature is equalized. In contrast, Old and Trevena [78] found that an alloy of zinc and tin can induce LME in steel. Landow et al. [79] however, state that, in general, alloys of lead with tin and also with bismuth can cause the LME phenomenon in a relatively wide temperature range from 230 to 470 °C.

The LME phenomenon itself is not the result of the interaction of only a specific metal alloy in the liquid state, but its course also requires the presence of tensile stresses in the material and the favorable chemical composition of the steel [23]. The appearance of new baths and increased cracking of products during galvanizing caused numerous discussions and studies on the impact of bath alloying additives on the possibility of causing the LME phenomenon. Poag and Zervodius [80] claim that lead enhances the propensity of liquid zinc to induce LME, while nickel has no effect on this phenomenon. On the other hand, Bi and Sn have no effect up to the content of 0.1 wt % and 0.2 wt %, respectively. However, above these levels, they can induce LME phenomena. The research showed [16,75] that lead, bismuth and tin accumulate in the crack tip, and their amount may be several times greater than their concentration in the bath. This leads to the formation of the alloy of these metals at the crack tip and an interaction as described by Landow [79]. In addition, studies were also conducted to determine the relationship between the chemical composition of steel and the structure of the product itself, and the risk of LME [81]. In order to limit this phenomenon, many guidelines included in the DASt-Richtlinie 022 directive developed in Germany [82] were indicated. The directive regulates all aspects of planning, construction, production and galvanizing. First of all, the restrictions concerned the composition of the galvanizing bath, and in particular, the reduction in the content of additives such as Bi, Sn and Pb in the bath. The content of alloying additives in the bath specified in the DASt-Richtlinie 022 directive in the case of galvanizing of load-bearing structures is presented in Table 2.

The DASt-Richtlinie 022 directive limits the Sn and Bi content in the bath much more strictly than the Pb content. Hence, many manufacturers continue to use lead despite its toxicity. The DASt-Richtlinie 022 directive was developed by Deutscher Ausschuß für Stahlbau and is generally applicable in Germany. However, its recommendations are also used by many galvanizing plants in Europe. However, it should be noted that the problem of cracking of products during galvanizing applies to a certain group of products. The galvanizing plant individually analyzes the range of galvanized products and adjusts the chemical composition of the bath to its own needs.

## 7. Conclusions

Maintaining the competitiveness of the zinc coating obtained by the batch hot-dip galvanizing method in relation to other anticorrosive protections determines the constant reduction in the coating production costs while maintaining its high quality. The galvanizing process, compared to other manufacturing processes receiving zinc and zinc alloys, generates the highest losses of zinc resulting from the production in zinc-rich waste (hard zinc, galvanizing ashes), the formation of economically unjustified, excessively thick coatings caused by the reactivity of steel and insufficient drainage of zinc from the surface of the product. Rational conduct of the galvanizing process requires reducing the amount of waste, and at the same time, to obtain high-quality coatings, it is important to control the reactivity of the steel. Such possibilities, apart from the appropriate process parameters, are provided by the appropriate selection of the chemical composition of the galvanizing bath. Over the last 40 years, dynamic development in the galvanizing technology has been observed in order to develop the best configuration of alloying additives for the zinc bath. Many alloying additives were tested and the ranges of their concentrations in the bath were determined. The reactivity of the steel in the Sandelin range can be controlled by adding Ni. Additionally, the control of the reactivity of the steel can be assisted by Al, Bi and Sn. The addition of Pb, Bi and Sn influences the appearance of the outer layer of the coating, further improving the drainage of liquid zinc from the surface of the product. Aluminum, in turn, protects the zinc bath against excessive oxidation. When selecting alloying additives, one should also take into account the negative effects of their impact, mainly lead toxicity and the possibility of increasing the risk of LME by Bi and Sn. Quantitatively, the rules of minimum and maximum bath concentration should be followed. Table 3 presents the most optimal configurations of the chemical composition of galvanizing baths currently used.

Qualitatively, the chemical composition of the bath must be selected individually depending on the galvanized product range. When selecting the chemical composition of the bath, however, the justification for the use of Pb and the limitations of Bi and Sn resulting from the DASt 022 directive should be considered. With such management of the composition of the zinc bath, the maximum effectiveness of the impact of the alloying elements can be obtained.

The desire to reduce the consumption of zinc determines further studies of the chemical composition of the zinc to batch hot-dip galvanizing. The direction of development is the working out of baths allowing to increase the corrosion resistance of coatings. Such possibilities are provided by the use of a zinc–aluminum bath, as well as Zn-Al, additionally containing Mg. However, this requires the development of appropriate fluxes enabling the production in coatings in these baths by the batch hot-dip method. The development of a technology for producing coatings with increased corrosion resistance will reduce the thickness of the coating, which in turn will reduce the consumption of zinc.

## Figures and Tables

**Figure 1 materials-13-04168-f001:**
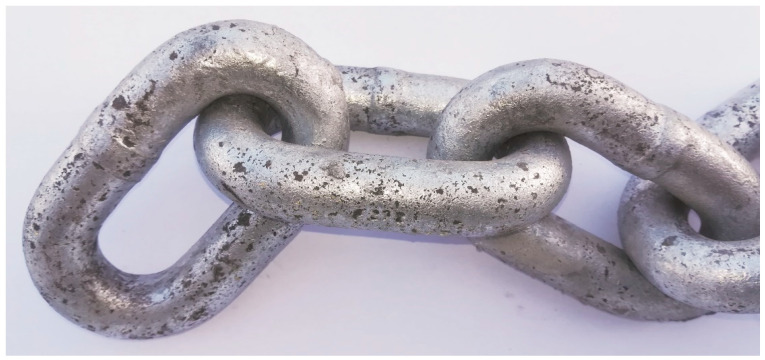
The appearance of the chain surface with a zinc coating obtained in a bath containing an excessive amount of Al (0.25 wt % Al); chain size 18 × 64 (photo by H.Kania).

**Figure 2 materials-13-04168-f002:**
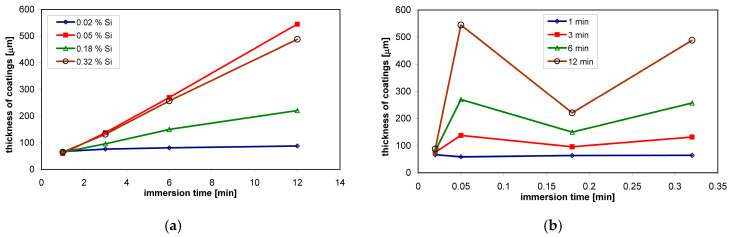
Changes in the thickness of the coating depending on: the time of immersion in the zinc bath (**a**) and the Si content in the steel (**b**).

**Figure 3 materials-13-04168-f003:**
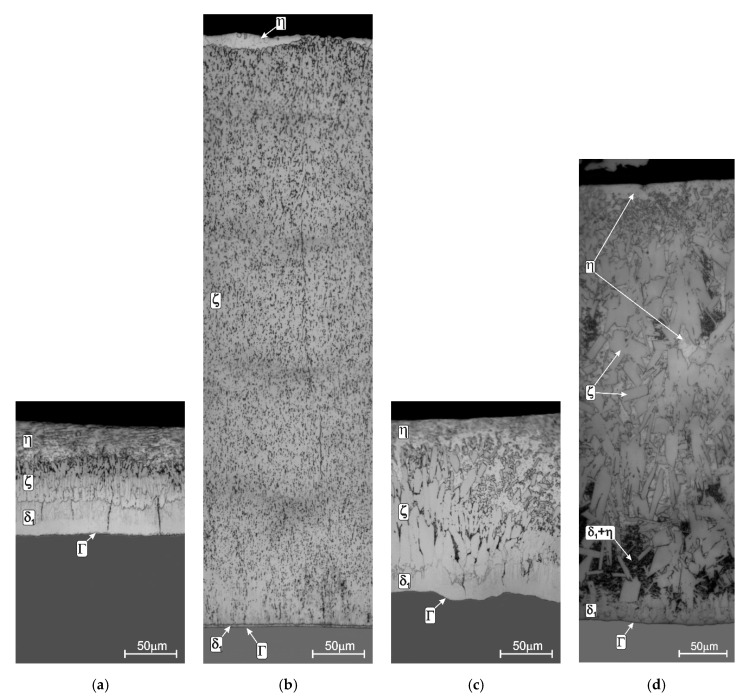
Structure of coatings obtained in the “pure” zinc bath on steels with different silicon content: low-silicon steel (**a**), Sandelin’s steel (**b**), Sebisty’s steel (**c**) and high-silicon steel (**d**); temperature: 450 °C, immersion time: 3 min, adapted from [36].

**Figure 4 materials-13-04168-f004:**
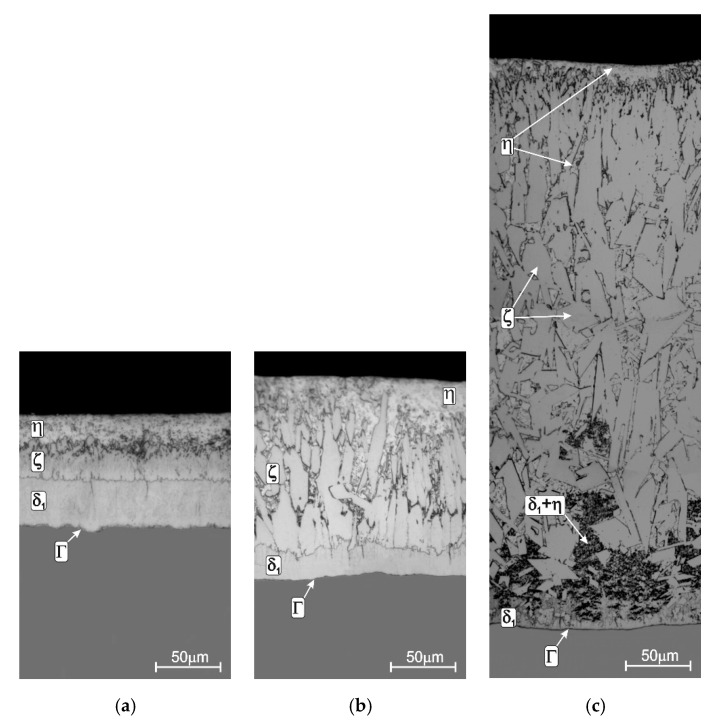
The structure of the coating obtained in the Zn + 0.05% Ni bath on: Sandelin’s steel (**a**), Sebisty’s steel (**b**) and high-silicon steel (**c**), adapted from [36].

**Figure 5 materials-13-04168-f005:**
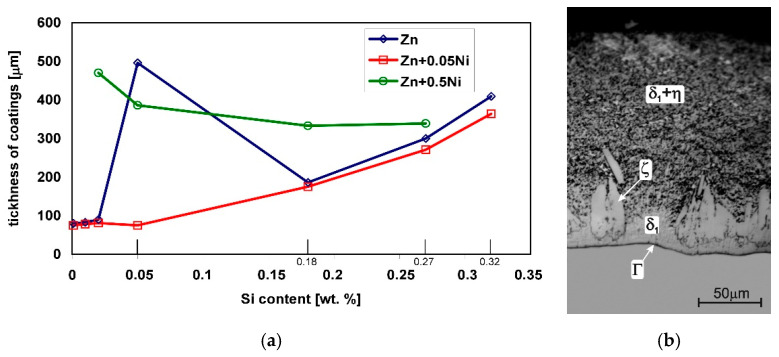
Comparison of changes in the thickness of coatings obtained in the Zn, Zn + 0.05% Ni and Zn + 0.5% Ni baths; data from [36] (**a**) and the structure of the coating obtained in the Zn + 0.5% Ni bath on high-silicon steel (**b**).

**Figure 6 materials-13-04168-f006:**
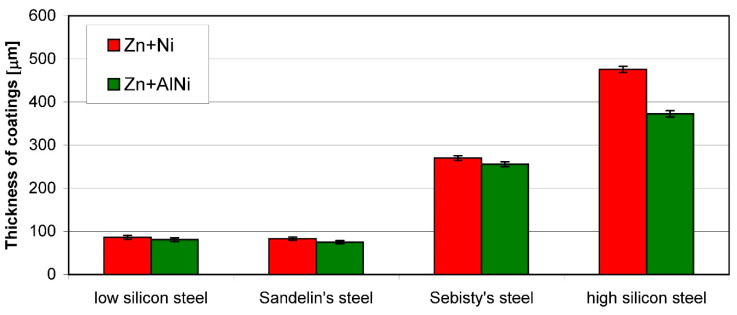
Influence of Al content (0.022 wt %) in Zn-Ni bath on the kinetics of coatings growth obtained on steels with different silicon content; temperature 450 °C, immersion time 12 min, selected data from [52].

**Figure 7 materials-13-04168-f007:**
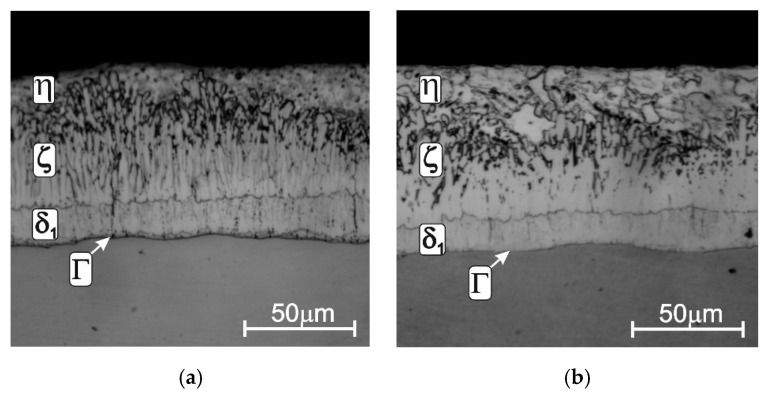
Structure of the coating obtained on Sandelin’s steel in Zn-Ni (**a**) and Zn-AlNi (**b**) bath, temperature 450 °C, immersion time 5 min, data from [53].

**Figure 8 materials-13-04168-f008:**
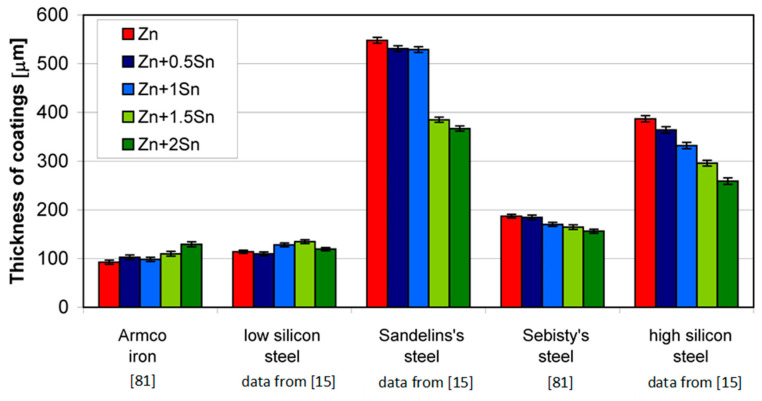
Thickness of coatings obtained in baths with the addition of tin; temperature 450 °C, immersion time 12 min.

**Figure 9 materials-13-04168-f009:**
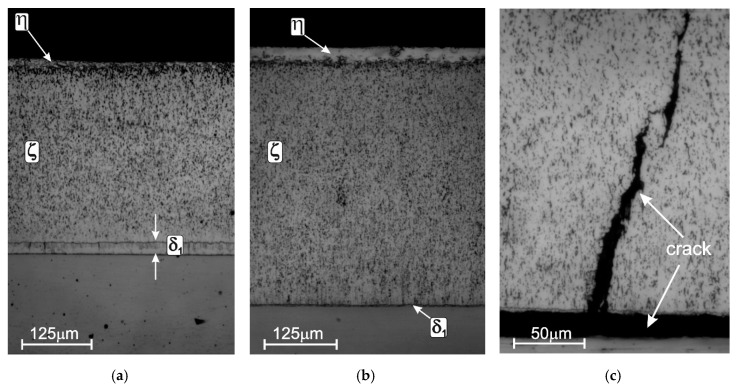
The structure of the coating obtained on Sandelin’s steel in the Zn + 2% Sn bath (**a**) [53], Sandelin’s effect in the “pure” zinc bath: excessive growth of the ζ phase (**b**) [53], and cracking and delaminating of the coating from the substrate (**c**) [63].

**Figure 10 materials-13-04168-f010:**
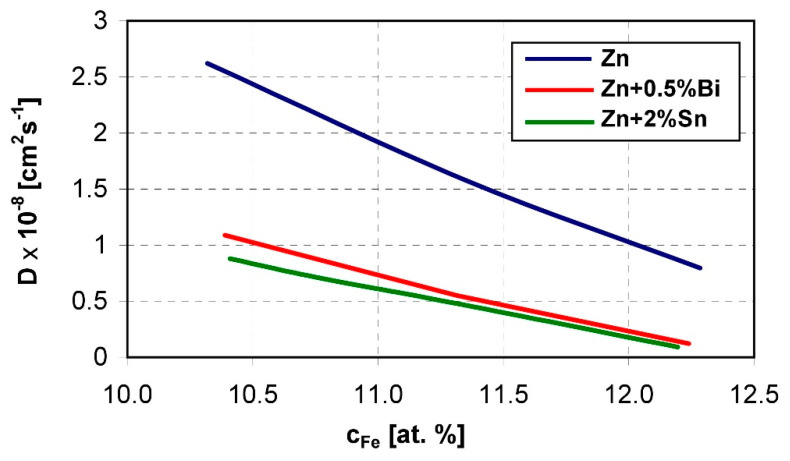
Dependence of the diffusion coefficient on iron concentration in the homogeneity of the δ_1_ phase during diffusion dissolution of Sandelin’s steel in “pure” zinc and in a zinc alloy with the addition of 2 wt %. Sn and 0.5 wt %. Bi; temperature 450 °C, [53].

**Figure 11 materials-13-04168-f011:**
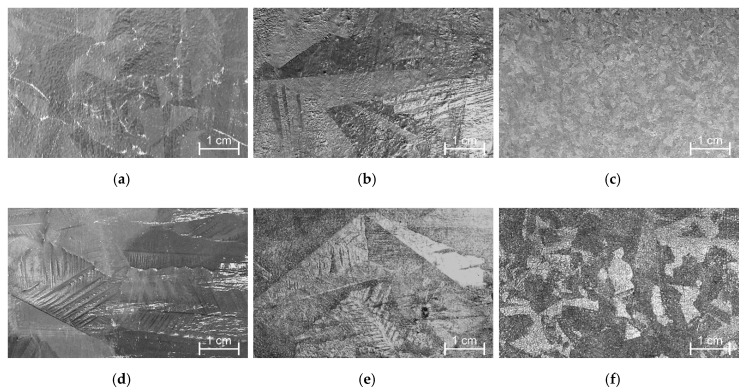
The appearance of the spangle obtained in a bath with 0.5 wt % Pb (**a**), 1 wt % Pb (**d**), 0.05 wt % Bi (**b**), 0.1 wt % Bi (**e**), 1 wt % Sn (**c**), 2 wt % Sn (**f**). The coating was obtained at the temperature of 450 °C with an immersion time of 3 min on a sample of low-silicon steel with dimensions of 100 × 50 × 2 mm, adapted from [36].

**Figure 12 materials-13-04168-f012:**
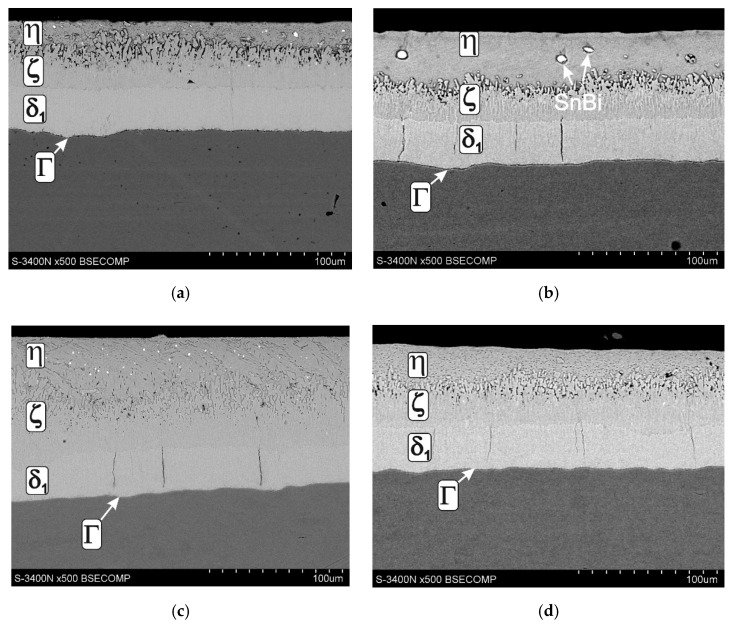
Microstructure of the coating obtained on: low-silicon steel in Zn-AlNiPb (**a**) bath, low-silicon steel in Zn-AlNiBiSn (**b**) bath, Sandelin’s steel in Zn-AlNiPb (**c**) bath, Sandelin’s steel in Zn-AlNiBiSn (**d**) bath.

**Figure 13 materials-13-04168-f013:**
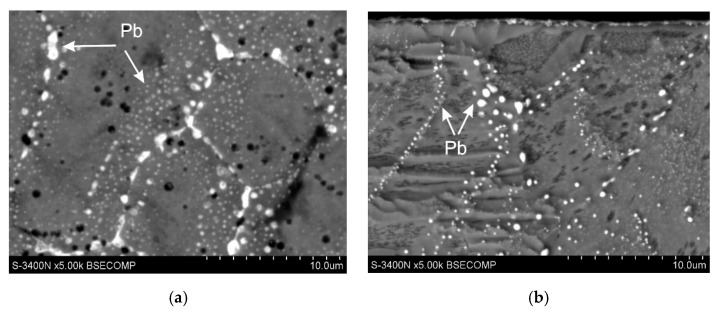
The precipitation of lead on the surface (**a**) and in the outer layer (**b**) of the coating obtained in the Zn-AlNiPb bath, adapted from [12] and (**c**) the precipitation of Sn and Bi in the diffusion layer of the coating obtained in the Zn-AlNiBiSn bath.

**Figure 14 materials-13-04168-f014:**
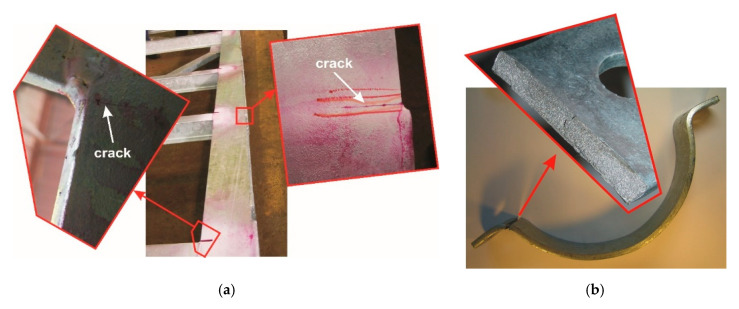
Cracks in galvanized products in the bath containing Bi and Sn as a result of the Liquid Metal Embrittlement (LME) phenomenon: welded supporting structure (**a**) and clamp for the swirl column holder (**b**) (photo by H.Kania)

**Table 1 materials-13-04168-t001:** Fe-Zn phase characteristic.

Phases	Formula	Content of Fe
Γ	Fe_3_Zn_10_	23.7–31.3 at % [37] 25.0–31.5 at % [38]
δ_1_	FeZn_10_	8.1–13 at % [37] 8.7–13.4 at % [39]
ζ	FeZn_13_	6.9–7.2 at % [37] 6.5–7.5 at % [39]
η(Zn)	Zn(Fe)	0.03 wt % [40]

**Table 2 materials-13-04168-t002:** The content of alloying additives in bath according to the DASt-Richtlinie 022 Directive; data from [82].

Sn	Pb + 10Bi	Ni	Al	Sum of the Remaining Elements (without Zn and Fe)
≤0.1 wt %	≤1.5 wt %	<0.1 wt %	<0.1 wt %	<0.1 wt %

**Table 3 materials-13-04168-t003:** Configurations of the most optimal hot-dip galvanizing baths.

Bath	Element Content, wt %				
	Al	Ni	Pb	Bi	Sn
Zn-AlNiPb	0.005–0.01	0.04–0.06	0.4–0.5		-
Zn-AlNiBi	0.005–0.01	0.04–0.06	-	0.05–0.1	-
Zn-AlNiBiSn	0.005–0.01	0.04–0.06	-	0.05–0.1	0.1–0.3

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
