# Peer review of "Development of Bath Chemical Composition for Batch Hot-Dip Galvanizing—A Review"

_materials, 2020, doi:10.3390/ma13184168_

Round 1
Reviewer 1 Report
On page 3 it is written : “In 1987 [4], Special High Grade (SGH) zinc with a high purity of 99.995% became the standard in the production of zinc, in which the level of all impurities does not exceed 0.005 wt.% [9]”. This is not correct. Reference 9 is a commercial presentation, it does not state the situation correctly. According to the 2016 IZA Zinc College lecture by Manuel Malaga, of the 111 active zinc smelters in the world, 88 are electrolytic, producing the Special High Grade Zinc and 23 are pyrolytic, producing a lower purity grade. Most of the lower purity grade is consumed by the general galvanizing lndustry. According to the American Galvanizers Association Process Survey of 2019, 10% of North American Galvanizers use GOB zinc, 4% use 99.9% High Grade Zinc and the rest use SHG or a mix of SHG and other zinc grades. The situation is similar in Japan. Therefore, it is not correct to write that SHG (not SGH as written) became the standard in 1987. It is correct to write that the use of Pb-containing galvanizing alloys has decreased over the years, and that many galvanizers have changed to SHG – but certainly not all.
On page 6 is the first of many references to the Zn-Fe compounds zeta, delta and gamma, but the composition of these phases is never mentioned. The composition of gamma-2 phase is written in the text. It could be instructive and efficient to include the Zn-Fe phase diagram, showing the location of the Zn-Fe phases and also the eta zinc solution which is also mentioned, its composition or identity is also not mentioned.
Author Response
Dear Reviewer,
We are grateful for taking your time to read our paper and for their constructive comments. We have carefully reviewed the comments and have revised the manuscript accordingly. Our responses are below given in a point-by-point manner. Changes to the text are shown in red in the revised manuscript. We hope the revised version is now suitable for publication.
On page 3 it is written : “In 1987 [4], Special High Grade (SGH) zinc with a high purity of 99.995% became the standard in the production of zinc, in which the level of all impurities does not exceed 0.005 wt.% [9]”. This is not correct. Reference 9 is a commercial presentation, it does not state the situation correctly. According to the 2016 IZA Zinc College lecture by Manuel Malaga, of the 111 active zinc smelters in the world, 88 are electrolytic, producing the Special High Grade Zinc and 23 are pyrolytic, producing a lower purity grade. Most of the lower purity grade is consumed by the general galvanizing lndustry. According to the American Galvanizers Association Process Survey of 2019, 10% of North American Galvanizers use GOB zinc, 4% use 99.9% High Grade Zinc and the rest use SHG or a mix of SHG and other zinc grades. The situation is similar in Japan. Therefore, it is not correct to write that SHG (not SGH as written) became the standard in 1987. It is correct to write that the use of Pb-containing galvanizing alloys has decreased over the years, and that many galvanizers have changed to SHG – but certainly not all.
We agree with the reviewer that indeed not all galvanizing plants in the world have completely abandoned the use of GOB zinc. Our statement was based on the knowledge of the European market, including mainly Germany (first in terms of production potential in Europe) and Poland (third in terms of production potential in Europe - data from EGGA), as well as our own information obtained from the Polish Galvanizing Association, Czech and Slovak Galvanizing Association, which are a member of the European General Galvanizers Association (EGGA). Line 114's claim is indeed inadequate. Therefore, we changed the text in Line 113-116.
We also apologize for the error in the marking of the Special High Grade (SHG). The error has been corrected.
We would also like to explain that References [9] is the European standard EN 1179 on the basis of which the chemical composition of SHG zinc was quoted.
At the same time, we thank you for pointing out a valuable source of information (IZA and AGA) on the production of zinc and its use by American galvanizers that was not available to us.
On page 6 is the first of many references to the Zn-Fe compounds zeta, delta and gamma, but the composition of these phases is never mentioned. The composition of gamma-2 phase is written in the text. It could be instructive and efficient to include the Zn-Fe phase diagram, showing the location of the Zn-Fe phases and also the eta zinc solution which is also mentioned, its composition or identity is also not mentioned.
We agree with the Reviewer that the text does not sufficiently characterize the phases of the Zn-Fe system. Because, due to copyright reasons, we do not want to copy the Zn-Fe equilibrium system, the description of the phases of this system has been supplemented and included in table 2.
English has been improved.
Reviewer 2 Report
The main remarks to presented work:
The title of the paper should be changed. Also, include hyphen before A review.
The “Introduction” is poorly made. I don’t see enough literature references here. It seems that most of the textbook information is shared here.
The review should include the overview of this manuscript or Table of contents in the last paragraph of the introduction section.
It would be better to summarize the various baths used in galvanization in the last few years in a Table format.
Why is this figure 1 here?
Why subfigures in figure 3 are of unequal size?
There should be standard error bars in coating thickness graphs (figure 6, figure 8).
I was surprised. There is no micron marker in figure 11. This is very unscientific.
Future directions should be given after conclusion section.
Author Response
Dear Reviewer,
We are grateful for taking your time to read our paper and for their constructive comments. We have carefully reviewed the comments and have revised the manuscript accordingly. Our responses are below given in a point-by-point manner. Changes to the text are shown in red in the revised manuscript. We hope the revised version is now suitable for publication.
The title of the paper should be changed. Also, include hyphen before A review.
We do not fully understand what the reviewer means when he says about changing the title. However, we can change the title to the following:
Development of bath chemical composition for batch hot dip galvanizing. A Review.
The form of "A Review." (without a hyphen) was used suggesting the titles of previous articles published in the Materials journal as a review.
The “Introduction” is poorly made. I don’t see enough literature references here. It seems that most of the textbook information is shared here.
In Introduction, literature references were supplemented.
The review should include the overview of this manuscript or Table of contents in the last paragraph of the introduction section.
Text in lines 90-95 has been supplemented.
It would be better to summarize the various baths used in galvanization in the last few years in a Table format.
The most rational and currently used solutions of chemical composition are summarized in Table 3. We believe that compiling both the current solutions and compositions already withdrawn in the form of a table may raise doubts as to their application.
Why is this figure 1 here?
Figure 1 shows an example of the appearance of a galvanized product in a bath with excessive Al content (0.25% Al). Indeed, the description may raise doubts, therefore the sentence in Line 171 and the caption under Fig. 1 were changed
Why subfigures in figure 3 are of unequal size?
Figure 3 shows the structure of coatings on steels with different Si content. To illustrate the differences and the influence of Si also on the coating thickness, the photos are shown at the same magnification (the same scale in the figures).
There should be standard error bars in coating thickness graphs (figure 6, figure 8).
The error bars are shown in Figs. 6 and 8.
I was surprised. There is no micron marker in figure 11. This is very unscientific.
The scale is now shown in fig. 11.
Future directions should be given after conclusion section.
Further directions of development and research on the hot-dip galvanizing process have been supplemented in Line 728-735
English has been improved.
Round 2
Reviewer 2 Report
The authors have done a good job in revising the manuscript. They have addressed all the comments.